# Nanolipogel Loaded with Tea Tree Oil for the Management of Burn: GC-MS Analysis, In Vitro and In Vivo Evaluation

**DOI:** 10.3390/molecules27196143

**Published:** 2022-09-20

**Authors:** Rabab Kamel, Sherif M. Afifi, Amr M. Abdou, Tuba Esatbeyoglu, Mona M. AbouSamra

**Affiliations:** 1Pharmaceutical Technology Department, National Research Centre, Cairo 12622, Egypt; 2Pharmacognosy Department, Faculty of Pharmacy, University of Sadat City, Sadat City 32897, Egypt; 3Department of Microbiology and Immunology, National Research Centre, Cairo 12622, Egypt; 4Institute of Food Science and Human Nutrition, Department of Food Development and Food Quality, Gottfried Wilhelm Leibniz University Hannover, Am Kleinen Felde 30, 30167 Hannover, Germany

**Keywords:** essential oil, lipid nanoparticles, cellulose nanofibers, lipogel, burn dressing, sesquiterpene, alcohols, dermal

## Abstract

The GC-MS analysis of tea tree oil (TTO) revealed 38 volatile components with sesquiterpene hydrocarbons (43.56%) and alcohols (41.03%) as major detected classes. TTO efficacy is masked by its hydrophobicity; nanoencapsulation can address this drawback. The results showed that TTO-loaded solid lipid nanoparticles (SLN1), composed of glyceryl monostearate (2% *w/w*) and Poloxamer188 (5% *w/w*), was spherical in shape with a core-shell microstructure. TTO-SLN1 showed a high entrapment efficiency (96.26 ± 2.3%), small particle size (235.0 ± 20.4 nm), low polydispersity index (0.31 ± 0.01), and high negative Zeta potential (−32 mV). Moreover, it exhibited a faster active agent release (almost complete within 4 h) compared to other formulated TTO-SLNs as well as the plain oil. TTO-SLN1 was then incorporated into cellulose nanofibers gel, isolated from sugarcane bagasse, to form the ‘TTO-loaded nanolipogel’ which had a shear-thinning behavior. Second-degree thermal injuries were induced in Wistar rats, then the burned skin areas were treated daily for 7 days with the TTO-loaded nanolipogel compared to the unmedicated nanolipogel, the TTO-loaded conventional gel, and the normal saline (control). The measurement of burn contraction proved that TTO-loaded nanolipogel exhibited a significantly accelerated skin healing, this was confirmed by histopathological examination as well as quantitative assessment of inflammatory infiltrate. This study highlighted the success of the proposed nanotechnology approach in improving the efficacy of TTO used for the repair of skin damage induced by burns.

## 1. Introduction

Burns are physical injuries that appear on the skin’s surface when exposed to hot bodies, sunlight, or chemicals [1]. These burns eventually lead to skin function impairment [2], resulting in bacterial infections and inflammations [3]. An ideal dressing can thus maintain high humidity levels at the wound site, it must be non-toxic, non-allergenic, comfortable, and cost-effective, allow for oxygen and water vapor exchange, and protect against microbial invasion [4]. Additionally, it must be highly biocompatible and have sufficient absorption capacity for burn exudates [5]. For this reason, the use of natural materials as a topical wound healing treatment has received special attention because of their safety and reduced cost in addition to their prospective efficacy [6,7]. Currently, many skin repair dressings have been incorporated with compounds of natural sources in order to enhance their bactericidal effect [8,9]. Essential oils are considered among these natural products displaying bactericidal activity that may be incorporated into dressings intended for skin repair [10,11].

Tea tree oil (TTO) is one of the essential oils isolated from *Melaleuca alternifolia* leaves used for burn wounds [12]. It has been reported that TTO showed to be effective against bacterial, viral, and fungal organisms and it is also a powerful immunostimulant agent. It helps the skin to heal by encouraging the formation of scar tissue [13]. For instance, hydrogel containing TTO reduced skin damage induced by ultraviolet B exposure and the hydrogel reduced wound area compared to allantoin-loaded hydrogel, posing TTO as a possible candidate to control skin inflammation and wounds [14]. TTO has also exhibited a topical antimicrobial effect against various fungi, bacteria, and yeasts [15,16]. TTO has more than 100 components, with monoterpene and sesquiterpene hydrocarbons, along with their associated alcohols, being abundant classes. The major constituents of TTO, i.e., erpinene-4-ol, *α*-terpineol, terpinolene, and terpinene, have been separately correlated to anti-inflammatory and antibacterial effects. Terpinen-4-ol was found to be the greatest contributor to antibacterial efficacy against *Staphylococcus aureus* and *Propionibacterium acnes* [17,18]. The most abundant component of TTO, terpinen-4-ol, has been shown to exert significant anti-inflammatory action via inhibiting the formation of TNF-*α* (tumor necrosis factor-*α*), prostaglandin E2, IL-1 (interleukin-1), IL-8, and IL-10 [19]. Further, the cell membrane of *Botrytis cinerea* was found to be the primary target of terpinen-4-ol indicative of TTO antifungal mode of action causing cell membrane leakage [20].

Unfortunately, skin irritation can limit its direct topical application as a wound-healing material. Moreover, the lipophilic nature of the oil can decrease its efficacy by diminishing its absorption through biological membranes and its greasiness reduces patient acceptability. Previous studies have reported that the nanoencapsulation of natural oils resolves these problems and provides many advantages [21,22,23,24].

The use of lipid nanocarriers is considered as one of the successful solutions to deliver the oil easily and safely. In addition, they are suitable carriers for application to damaged and inflamed skin and they have a well-reported occlusive property based on their lipid nature, an important factor in accelerating wound healing [25]. Some previous publications have proven the efficiency of lipid-type nanocarriers for dermal delivery and repair [26,27,28,29]. These studies were focused on enhancing the solubility and efficacy of natural compounds with low aqueous solubility (such as rutin, diosmin, resveratrol, curcumin, and lutein) to target their delivery efficiently to the skin. The results proved the possible amelioration of their effects by nanoencapsulation in lipid-based colloidal carriers. On the other hand, it was proved that cellulose nanofibers (CNF) can significantly enhance cell regeneration and tissue healing [30,31,32]; also, they have a high water-holding capacity and, consequently, a gel-like appearance even at low concentrations [33], hence they can be used as a bioactive matrix for skin dressing.

GC-MS is increasingly being utilized to evaluate the purity of natural and pharmaceutical products [34]. The present study tackles the ability of solid lipid nanoparticles (SLNs) to encapsulate TTO (tea tree oil) in light of its chemical attributes, using GC-MS. After physicochemical investigations, the selected formulation was incorporated into cellulose nanofiber (CNF) gel to form TTO-loaded nanolipogel for dermal application in rat skin burn model.

## 2. Results and Discussion

### 2.1. GC-MS Analysis

GC-MS analysis has been extensively applied to analyze specimens containing essential oils [35]. A total of 38 volatile components have been detected in GC-MS chromatographic profile (Figure 1) belonging to various classes, i.e., alcohol, aldehyde, monoterpene and sesquiterpene hydrocarbons, as listed in Table 1. The major detected classes were sesquiterpene hydrocarbons and alcohols amounting at 43.56 and 41.03%, respectively, and in agreement with previous work [36] confirming the superb antioxidant and antimicrobial activities of tea tree oil. *α*-Farnesene (peak no. 23), the major sesquiterpene hydrocarbon, amounted to 9.43%, had a sweet apple fragrance [37], and was reported as an activator for antibacterial-related factors [38]. *α*-Cubebene (22), detected at 8.15%, contributed to the antioxidant and anti-inflammatory effects of the oil [39]. *α*-Bergamotene (18), another sesquiterpene hydrocarbon, constituted 5.3% and presented antibacterial activity [40]. Other prevalent sesquiterpene hydrocarbons included *α*-copaene (10, 13, 24), *β*-caryophyllene (16, 17), and *δ*-cadinene (20, 21), exhibiting anticancer and antioxidant effects [41]. On the other hand, alcohols were the second major class, owing to the enrichment of oil with erpinene-4-ol (12.06%), spathulenol (5.8%), trans-ascaridol glycol (4.66%), and *α*-terpineol (4.43%). Terpinen-4-ol (4) exerted antibacterial and antifungal activities [42], while spathulenol (27, 33) demonstrated anti-hyperalgesic and anti-nociceptive effects in chronic pain [43]. trans-Ascaridol glycol (8) was previously reported in oils of Chenopodium, revealing antimicrobial and anti-inflammatory activities [44]. *α*-Terpineol (5) had antifungal, antioxidant, anti-inflammatory, and antibacterial properties [45]. *α*-Terpinolene (2), the major monoterpene hydrocarbon detected at 5.27%, showed antibacterial activity [46]. The ability of the oil to control and cure skin burns can be expected due to its chemical constitution.

### 2.2. Rational for the Selection of the SLN Components

The SLNs under investigation were prepared using different concentrations of the lipid (glyceryl monostearate) and the surfactant (Poloxamer 188).

Poloxamer 188 (P188) is a non-ionic surfactant composed of polyoxyethylene/polyoxypropylene chains and it has a high HLB value (29) [47]. It was previously used for the preparation of Resveratrol-loaded lipid colloidal carriers for dermal delivery and showed promising results [48]. Some previous studies reported that a surfactant with a highHLB value resulted in the increment of the active ingredient’s entrapment due to its aqueous solubilization which allows its accommodation at the lipid/water interface and permits the creation of a shell enriched with the active agent around the lipid core [49,50].

Glyceryl monostearate (GMS) has previously been used as a lipid matrix to formulate SLN incorporating a lipophilic active agent of natural origin similar to TTO [51]. It is widely used as an emulsifier in cosmetics and pharmaceutical formulations and for the preparation of lipid nanoparticles [52]. Furthermore, it was reported that it improved skin penetration due to its lamellar structure similar to that of the human stratum corneum [53,54].

### 2.3. Physicochemical Properties of Prepared TTO-Loaded SLN

#### 2.3.1. Entrapment Efficiency and Loading Capacity

Results illustrated in Table 2 clearly show the high entrapment efficiency of SLNs (above 96%) which may be related to the synergistic emulsifying properties of both GMS and P188 which may lead to increase solubility of the TTO in the lipid and consequently increased the EE%, in addition to the lipophilic nature of the oil. The increase in GMS concentration afforded more space to encapsulate more drugs which reduces the drug partition in the outer phase [55]. However, increasing P188 concentration from 5% to 10% (*w/w*) while keeping the GMS concentration constant did not significantly affect the EE%; this may be due to its high HLB value (HLB = 29). On the other hand, TTO-SLN1 attained the significantly highest loading capacity (29.81 ± 1.2) compared to other SLNs.

#### 2.3.2. Particle Size Analysis (Mean Particle Size, Polydispersity Index, and Zeta Potential)

The mean particle size, polydispersity index, and zeta potential of SLNs are important characteristics from which their stability can be predicted.

The results listed in Table 2 reveal that all the SLNs are in the nanometric size, they range from 226.9 ± 20.6 to 429.8 ± 35.8 nm. Some previous studies have shown that a particle size of around these values was beneficial for dermal use and skin-targeting allowing for the accumulation of the active agent in the skin layers [26,27,56,57]. The results revealed that the particle size increased significantly with increasing the lipid (GMS) concentration from 2% to 4% and 8% *w/w* (*p* < 0.05); this could be explained by the fact that the homogenization efficiency decreases with increasing the content of the dispersed phase (lipid phase) [58]. On the other hand, increasing the concentration of P188 did not affect the particle size.

The polydispersity index is a ratio providing information about the homogeneity of particle size distribution in the system. As depicted in the results, the PDI listed in Table 2 ranged between 0.31 ± 0.01 and 0.44 ± 0.02, indicating a narrow and homogenous particle size distribution [59]—the higher the PDI, the lower the formulation uniformity [60]. Additionally, it was reported that the dynamic light scattering analysis was suitable for the measurement of nanoparticles with PDI values of less than 0.7 [61].

The zeta potential is a key factor in evaluating the stability of colloidal dispersion. The zeta potential values of different TTO-SLNs formulations are shown in Table 2, the results showed that they possess a good stability and dispersion quality. It was reported that nanoparticles with zeta potential values above +30 mV or below −30 mV are thermodynamically stable [62] and that P188 provides a steric hindrance for maintaining the stability of SLNs [63].

#### 2.3.3. In Vitro Release Studies

The release of the active agents from the prepared TTO-SLNs nanoformulations was compared to that of the plain TTO; the results illustrated in Figure 2 reveal a significant increase of the amount released in case of the prepared SLNs (*p* < 0.05). This can be explained by the successful solubilization and nano-encapsulation of the hydrophobic oil. At the same time, the hot homogenization technique plays a role in increasing the amount of TTO released through increasing the amount of oil partitioning on the outer shell and consequently causing a relatively rapid release [64]. TTO-SLN1 and TTO-SLN2 attained the fastest release, which was almost complete after 4 h, and this can be due to their smaller particle size compared to the other TTO-SLNs which resulted in a larger surface area available for active agent release. Furthermore, the lower concentration of the lipid (2% *w/w*) in TTO-SLN1 and TTO-SLN2 compared to the other TTO-SLNs led to the higher hydrophilicity of the formed nanoparticles and, hence, the greater release. On the other hand, a biphasic release pattern can be seen; this was previously attributed to the core-shell structure of the lipid-type nanoparticles and it allows for a fast release from the outer shell, followed by a lower rate from the inside core [48].

Based on the findings displayed above, TTO-SLN1 was selected to continue the study as it had the most suitable physicochemical properties (high entrapment efficiency (96.26 ± 2.3), small particle size (235 ± 20.4 nm), low PDI (0.31 ± 0.01), high ZP (−32 mV), fastest release, and the highest loading capacity when prepared using the lowest surfactant concentration (5% *w/w*).

### 2.4. Characterization of Optimized TTO-SLN

#### 2.4.1. Transmission Electron Microscopy (TEM)

The morphology of TTO-SLN1 is illustrated in Figure 3. As depicted in the figure, the TEM study shows that the particles are spherical, well-separated and uniform, however, their size is smaller than that recorded by the Zetasizer, which is based on dynamic light scattering because these measurements are performed using an aqueous dispersion of the nanoformulation; then, drying is performed before the TEM study [65]. The phenomenon of nanoparticle shrinkage was previously noted for lipid nanoparticles and was explained by the sample preparation method being used for high-resolution microscopy technique leading to particles shrinkage [66]. Furthermore, the micrograph shows the core-shell structural model commonly observed for solid lipid nanoparticles [48].

#### 2.4.2. Fourier-Transform Infrared Spectroscopy (FTIR)

The FTIR spectra of TTO, P188, GMS, and TTO-SLN1 are presented in Figure 4. TTO exhibits a stretching vibration peak corresponding to the C–H bond appeared at 2965 cm^−1^
Figure 4A [67]. As demonstrated in Figure 4B, a specific peak of P188 could be seen at 2860 cm^−1^ [68]. Additionally, the peaks corresponding to C-O stretching vibration were obtained at 1087.07 and 1024.20 cm^−1^ [69]. The GMS spectrum exhibited characteristic peaks at approximately 1471 cm^−1^ (CH3 bending), 1732 cm^−1^ (CO stretching), 2910 and 2848 cm^−1^ (CH stretching), and 3313 cm^−1^ (OH stretching) [70] (Figure 4C). The shifting and modification of the characteristic peaks of the individual components in the spectrum of TTO-SLN1 (Figure 4D) prove the incorporation of the ingredients within the newly formed nanostructure.

### 2.5. Evaluation of TTO-Loaded Nanolipogel

The prepared TTO-SLN/CNF gel (TTO-nanolipogel) was evaluated as follows:

#### 2.5.1. Determination of Gel pH

The pH of the TTO-nanolipogel was 6.2 ± 0.028. This value was compatible with the skin pH which ranges from 5.0 to 7.0 [71], and thus could be safely applied to the skin.

#### 2.5.2. Rheological Properties

The determination of gel strength requires the measurement of rheological properties [72]. The rheogram (Figure 5A) proves a non-Newtonian shear thinning flow which is defined by a decrease in gel viscosity as the shear rate increases due to the occurring polymeric network break down and was previously reported for cellulose nanofibers gel [73]. This is a desirable property for pharmaceutical preparations used topically to facilitate their application specially is the case of skin damage occurring in burns.

#### 2.5.3. In Vitro Release Studies for TTO-Nanolipogel

The in vitro release profile of the active agents from TTO-nanolipogel was compared to that of TTO-SLN1 (Figure 5B), the release was extended over 6 h. For the former, this can be due to the entrapment of the TTO-SLN within the polymeric fibrous network of cellulose.

### 2.6. In Vivo Studies

Burns are severe injuries that result in the destruction or breaking of the skin and are sometimes linked to a high death and morbidity rate [74]. When a new epithelium grows over the damage, as seen by visual inspection, the burn is deemed healed [75]. Due to their high cost and potential adverse effects such as allergy or medication resistance, synthetic medicines used to treat burn injuries are currently limited [76]. As a result, these problems must be addressed by developing new treatment strategy based on naturally derived ingredients that can promote wound healing while simultaneously meeting therapeutic standards. For controlling burns, a rapid onset of action is needed, therefore, the fast-releasing formulation (TTO-SLN1) is selected. After active agents’ release to the skin surface, then an extended effect is expected due to the accumulation of the lipid-type nanoformulation in the skin. It was previously reported that lipid-based nanocarriers with a particle size range around the value listed above (Table 2) was beneficial for skin targeting, permitting for the deposition of the active agent within the skin layers [26,27,56,57]. A once-daily dermal application is generally followed when using such a type of nanocarriers [56]. Additionally, this rate of application seems to be practical in case of burn dressing. Moreover, to facilitate the process of application in addition to taking advantage of the cell regeneration and tissue healing properties of the CNF, the formula was incorporated into the gel to form the TTO-loaded nanolipogel.

#### 2.6.1. Skin Irritation Test

The skin irritancy of the TTO-loaded nanolipogel as well as the plain oil was assessed visually in comparison to the negative control. The former was a non-irritant (PII = 0.67 ± 0.12), while the plain oil was an irritant (PII = 2.66 ± 0.57), which shows the importance of incorporating TTO in the suggested nanocarrier for a safe and comfortable topical application.

#### 2.6.2. Burn Healing

To assess the degree of burn healing in rats, color images of lesions were obtained on days 2, 4, 6, and 7 after burn induction, using a digital camera (Figure 6). In the first day of burn induction, visual observations revealed a comparable appearance of burn areas with a typical dark red coloration indicating the formation of a blood clot. The clot began to transform into scab on the fourth day. Retraction of the burn area was visible on days 6 and 7, with higher retraction in the burn sites treated with TTO-loaded nanolipogel followed by those treated with the TTO-loaded conventional gel. It was clear that the applied TTO-loaded nanolipogel (area A) significantly accelerated the process of burn healing when compared with the normal saline (area D), the unmedicated nanolipogel (area B), or the TTO-loaded conventional gel (area C). Moreover, some improvement of burn regions was demonstrated by the latter. All treatments were better than the normal saline.

To evaluate the percentage of burn healing in each group, a digital camera was placed at a fixed distance from the rats to take digital images of the burn areas on days 2, 4, 6, and 7 after the induction of the burn. The burn areas in each group were then measured using the Fiji image processing software. The percentage of burn contraction was then calculated (Table 3) based on the following equation [77]: X = [(A2 − Ax)/A2] × 100
where A2 is the surface area of the burn on the second day and Ax is the surface area of the burn on day x.

It is clear that the burn contraction was in the following order: A > C > B > D, treatment with the TTO-loaded nanolipogel attained the significantly fastest and best healing compared with other treatments (*p* < 0.05); this was expected as a result of the nanoencapsulation of TTO in the lipid-type nanoformula enhancing the release of the active agents and facilitating their penetration and deposition into the skin layers, which in turn accelerated the healing process and dermal repair [26,27,28,29,56]. The partial healing effect of the unmedicated lipogel may be due to the well-documented proliferative and regenerative properties of cellulose nanofibers matrix [30,32,78].

### 2.7. Histopathological Observations

Histopathological examination confirmed the visual observation and the calculated percentage of the burn area contraction. The burned areas treated with normal saline (negative control) showed acanthosis in the epidermis, while the underlying dermis showed granulation tissue formation and inflammatory cell infiltration with oedema (Figure 7a). When the burn was treated daily with TTO-loaded conventional gel, there was acanthosis in the epidermis associated with granulation tissue formation and few inflammatory cells’ infiltration in the dermis (Figure 7b). Whereas, for the burn area treated with the unmedicated nanolipogel, focal ulceration with acanthosis in the adjacent area was detected in the epidermis associated with granulation tissue formation and inflammatory cell infiltration in the underlying dermis (Figure 7c). The burn area treated with TTO-loaded nanolipogel showed a well-formed basal cell layer of the epidermis, as well as neovascularization and condensed dermal connective tissue beneath the basal cell layer, as seen without histopathological alteration (Figure 7d). The histopathological examination agreed with the results recorded by the semi-quantitative assessment of inflammatory infiltrate scored in the examined sections (Figure 8).

## 3. Materials and Methods

### 3.1. Materials

Tea tree oil (TTO) was purchased from Nefertari Health Care, Egypt. Kolliphor P188 (P188) (triblock copolymer of polyoxyethylene-polyoxypropylene, M.W. 162.23), glyceryl monostearte (GMS) (1-Stearoyl-*rac*-glycerol, M.W. 358.56), cellulose membrane (molecular weight cut-off 12,000–14,000 Da) were purchased from Sigma Aldrich (Burlington, MA, USA). Carboxymethyl cellulose (CMC) and all other reagents were obtained from El-Nasr Company for Pharmaceutical Chemicals (Cairo, Egypt).

### 3.2. Methods

#### 3.2.1. GC-MS Analysis

Five microliters of tea tree oil diluted in *n*-hexane (1 mL) was spiked with internal standard—10 μg of (*Z*)-3-hexenyl acetate. GC-MS analysis was conducted as previously reported [79]. An Agilent 7890 GC apparatus coupled with Agilent J&W HP-1 capillary column (60 m length, 0.25 mm i.d., and 0.25 μm film thickness), equipped with an Agilent 7693 auto-sampler and a quadrupole mass spectrometer, was used. For tea tree oil specimen, three different replicates were examined for the evaluation of biological variance under identical circumstances; the oven was maintained for 3 min at 40 °C, then programmed at 12 °C/min to 180 °C. For the purpose of cleaning the column, the post run was adjusted at 240 °C and maintained for 5 min. The spectra from *m/z* 40 to 500 were captured at 70 eV for the scan measurements. Utilizing a probability-based matching method, the spectra were compared to the databases (Wiley and NIST) for compound identification. Relative retention indices were used to further identify the volatiles by comparing them to both published research and commercially available reference standards.

#### 3.2.2. Preparation of Oil-Loaded Solid Lipid Nanoparticles (SLNs)

Glyceryl monostearate (GMS) was melted at 60 °C, then TTO was dissolved in it after cooling to 40 °C. The lipid phase containing the oil was poured into a hot aqueous surfactant solution of poloxamer 188 (P188). The mixtures were homogenized at a stirring speed of 25,000 rpm for 5 min using Heidolph Homogenizer (Silent Crusher Homogenizer, Germany). The obtained O/W emulsions were sonicated for 30 min at 40 °C (two cycles of 15 min with a 15 s interval) [80,81]. Finally, the preparations were cooled down in the ambient temperature to form the SLNs [82,83,84]. Different lipid-to-surfactant ratios were studied to evaluate their effect on the physical properties of the prepared SLNs. The detailed composition of the formulations is presented in Table 4.

Tea tree oil (TTO) 0.2% *w/w*, Glyceryl monostearate (GMS), and Poloxamer 188 (P188). The batch size was 25 g.

#### 3.2.3. Encapsulation Efficiency and Loading Capacity

For the assessment of the entrapment efficiency of TTO within the SLNs, the amount of the encapsulated TTO was measured indirectly by determining the amount of unentrapped TTO in the supernatant after centrifugation [85]. Centrifugation of SLNs was performed at 9000 rpm for 30 min at −4 °C. Then, the supernatant was collected and filtered through a Millipore membrane filter (0.2 µm), then diluted with ethanol and measured spectrophotometrically at 264.4 nm. The entrapment efficiency was calculated as follows:$$E.E.\;\%\;=\;\frac{W\;\mathrm{initial}\;\mathrm{TTO}\;\mathrm{added}-W\;\mathrm{TTO}\;\mathrm{free}}{W\;\mathrm{initial}\;\mathrm{TTO}\;\mathrm{added}}\;\times \;100$$

After centrifugation, the obtained residue (SLN) was collected, dried and the TTO loading was calculated according to the following equation [86,87]:
$$\mathrm{LC}\%=\frac{\mathrm{Weight}\;\mathrm{of}\;\mathrm{the}\;\mathrm{entrapped}\;\mathrm{TTO}}{\mathrm{Weight}\;\mathrm{of}\;\mathrm{the}\;\mathrm{obtained}\;\mathrm{SLN}}\times 100$$

#### 3.2.4. Examination of Physicochemical Properties of Nanoparticles: Particle Size, Zeta Potential, and Polydispersity Index

Zetasizer (Zetasizer Nano ZS; Malvern) was used to measure the particle size (PS), zeta potential (Z), and polydispersity index (PDI), using the dynamic light scattering technique. To achieve the aforementioned properties, the equipment parameters were adjusted, and the formulations were diluted 1:10 with double-distilled water before analysis.

#### 3.2.5. In Vitro Release Study

The in vitro release study was performed using the dialysis bag diffusion technique [88]. The dialysis bags (molecular weight cut off 12,000–14,000) were soaked in deionized water for 12 h before use. The plain oil (1 mg) or the SLNs dispersions (containing an equivalent amount of the oil) were filled into the bag (12.5 cm^2^) and tied at both ends and placed in a beaker containing 100 mL of 70% ethanolic aqueous solution [24,89]; temperature and speed were maintained at 32 ± 1 °C and 100 rpm, respectively. The samples were withdrawn at predetermined time intervals, and the same volume was replaced with fresh medium to maintain the sink condition. The samples were analyzed at 264.4 nm spectrophotometrically. The cumulative percent of released active agents was plotted against time.

#### 3.2.6. Characterization of Optimized TTO-SLN

##### Transmission Electron Microscopy

The morphology of the selected TTO-loaded SLN was examined using TEM (JEOL, JEM-1230, Tokyo, Japan). One drop of the diluted sample was placed on carbon-coated grids with films for examination.

##### Fourier-Transform Infrared Spectroscopy (FTIR)

The infrared spectra of the selected TTO-SLN as well as its individual components were determined as KBr discs using a Shimadzu 435 U-04 IR spectrophotometer with a spectrum range between 400 cm^−1^ and 4000 cm^−1^.

##### Preparation of TTO-Loaded Nanolipogel

Cellulose nanofiber (CNF) gel originating from sugarcane bagasse previously prepared and characterized by Kamel et al., 2020 [31] was used for the preparation of the TTO-loaded nanolipogel. The CNF gel was mixed with the TTO-loaded SLN in a ratio of 1:1 at room temperature, using a magnetic stirrer until homogeneity.

#### 3.2.7. Characterization of TTO-Loaded Nanolipogel

##### Determination of pH

TTO-loaded nanolipogel (0.5 g) was dispersed in 10 mL distilled water. The pH values were recorded using a digital pH meter (Jenway, Bibby Scientific Limited, Staffordshire, UK) at 25 °C. The pH meter was first standardized using buffer solution at pH 7.0 and pH 10.0. All measurements were performed in triplicate.

##### Rheological Study

In order to determine the rheological behavior of the TTO-loaded nanolipogel, the flow curve of prepared gel was obtained using a computerized rheometer (Anton Paar, Physica MCR 301) equipped with a cone and plate at 37 °C (plate diameter—40 mm, cone angle—4°). Continuous variation of the speed rate for each sample (from 0–300 s^−1^) was applied and the resulting viscosity was recorded.

##### In Vitro Release Study

The in vitro release from the TTO-loaded nanolipogel compared to the TTO-SLN (containing an equivalent amount of TTO) was evaluated using the dialysis bag diffusion technique, following the procedure described above.

### 3.3. In Vivo Studies

#### 3.3.1. Animals

Adult albino Wistar rats, weighing 200–250 g, were obtained and kept in the animal house unit of the National Research Centre, for at least one week prior to the experiments, under standard conditions of light and temperature. All animals had access to a standard laboratory diet consisting of vitamin mixture (1%), mineral mixture (4%), corn oil (10%), sucrose (20%), cellulose (0.2%), casein pure (10.5%), and starch (54.3%). Food and water were supplied ad libitum throughout the duration of study.

#### 3.3.2. Skin Irritation Test

The irritancy of the prepared TTO-loaded nanolipogel was compared to that of the plain oil based on a previously modified Draize test [90,91,92]. Briefly, 24 h after shaving the dorsal side of the rats, they were divided into three groups: the normal control and those receiving the TTO-loaded nanolipogel and the plain oil. The application was performed once daily for 3 consecutive days, then the application sites were visually examined for oedema and erythema at 24 and 72 h. Scoring (0–4) was performed as previously explained in detail; score 0 indicates no skin reactions, while score 4 indicates severe oedema and erythema [90,91,92,93]. The final score represents the average of the 24 and 72 h readings. The primary irritancy index (PII) was determined for each group by adding the oedema and erythema scores; the treatments were classified as non-irritant if PII < 2, irritant if PII = 2–5, and highly irritant if PII > 5.

#### 3.3.3. Induction of Burn Wounds

The induction of second-degree thermal injuries in the animals was carried out according to a previously published method [94] with minor modifications and the study was approved by the ethical committee of the National Research Centre (approval number 19281). In brief, the rats (*n* = 5) were anesthetized by an intraperitoneal injection of ketamine (75 mg/kg) and xylazine (10 mg/kg) and the hair of backside of the animals was shaved. Then, the shaved area was antisepticised with the 1% povidone-iodine. A solid aluminum bar (diameter = 10 mm) was heated to a temperature of nearly 97 °C and pressed to the shaved and disinfected back of the animals for 20 s to induce four burning positions (diameter of each burn = 10 mm) on the back of each rat. The four burned areas were treated daily with one of the following treatments (100 mg): A = TTO-loaded nanolipogel, B = unmedicated nanolipogel (without the oil), C = TTO-loaded conventional gel, and D = normal saline (reference group). All the treatments were applied, once a day, in a fine layer covering the surface of the burns using sterile cotton swab. An analgesic dipyrone sodium (SynZeal Inc., Gujarat, India) was administered (40 mg/kg, i.m.) to all animals after the induction of burns to prevent animals suffering. Then, each animal was placed in a separate cage until the end of the experiment to avoid licking or biting of wound areas by other animals. On days 2, 4, 6, and 7 after burn injury, color photographs of the wounds were taken by digital camera (Canon). In the last day (day 7) of the experiment, animal scarification was performed by decapitation under anesthesia and their debris were disposed according to the guidelines of National Research Centre.

TTO-loaded conventional gel was composed of carboxymethyl cellulose (CMC) gel (5% *w/w*), prepared as previously explained to be suitable to incorporate the lipophilic oil [95]. The gel was loaded with an equivalent amount of the oil as the TTO-loaded nanolipogel.

### 3.4. Histopathological Evaluation

Autopsy samples were taken from the skin of rats on day 7 in burn areas with normal tissue surrounding them and fixed in 10% formol saline for twenty-four hours. Washing was done in tap water, then, serial dilutions of alcohol (methyl, ethyl, and absolute ethyl) were used for dehydration. Specimens were cleared in xylene and embedded in paraffin at 56 °C for 24 h. Paraffin beeswax tissue blocks were prepared for sectioning at 4 µm thickness by LEITZ ROTARY microtome. The obtained tissue sections were collected on glass slides, deparaffinized, stained by hematoxylin and eosin stain for routine examination through the light electric microscope [96].

### 3.5. Assessment of Inflammation

Paraffinized sections (4 µm) from each rat were stained with hematoxylin and eosin. A semi-quantitative scale was used to assess the degree of inflammation through examination of the slides under a light microscope. Five different fields were counted, and the average was subsequently calculated [97]. The level of inflammation was determined by measuring the inflammatory infiltrate as follows: 0: Nil; 1: mild, 2: moderate, and 3: severe.

### 3.6. Statistical Analysis

Data analysis was performed based on one-way analysis of variance (ANOVA) followed by Tukey’s test for multiple comparisons. SPSS software was used. The difference was considered significant when *p* < 0.05.

## 4. Conclusions

GC-MS analysis of tea tree oil was performed, the major components detected have a synergistic effect on suppression of secondary microbial infection, removal of free radicals, and reduction of inflammatory mediators, which facilitates tissue healing. This study was focused on the nanoencapsulation of tea tree oil (TTO) in lipid-based nanoparticles to overcome its high lipophilicity. The selected TTO-loaded solid lipid nanoparticles (TTO-SLN) were further embedded in cellulose nanofibers gel to be applied topically as burn dressing. The results proved that the designed nanoformulation with naturally derived ingredients has the ability to accelerate the healing process, encourage cell proliferation, provide better re-epithelialization and can be a promising approach for the management of burns.

## Figures and Tables

**Figure 1 molecules-27-06143-f001:**
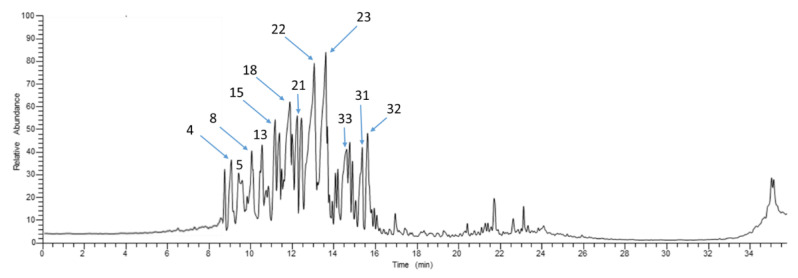
Representative total ion chromatograms of volatiles acquired from tea tree oil via GC-MS. For peak numbers, refer to Table 1.

**Figure 2 molecules-27-06143-f002:**
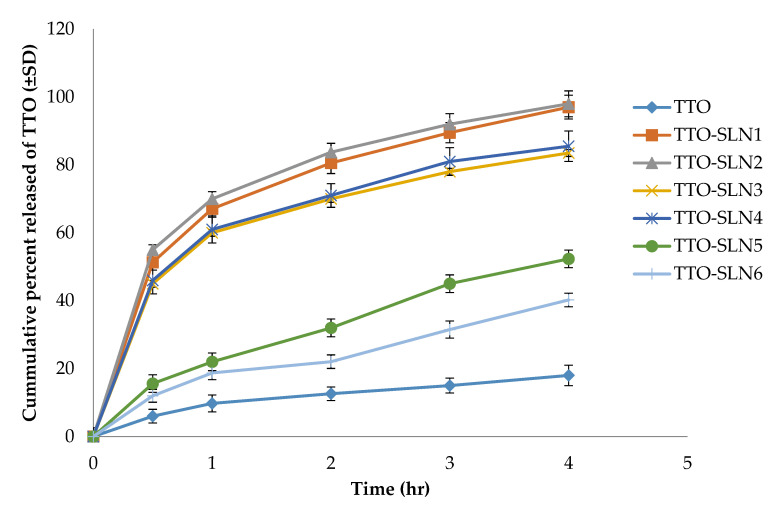
In vitro release study of the plain oil compared to the formulated SLNs.

**Figure 3 molecules-27-06143-f003:**
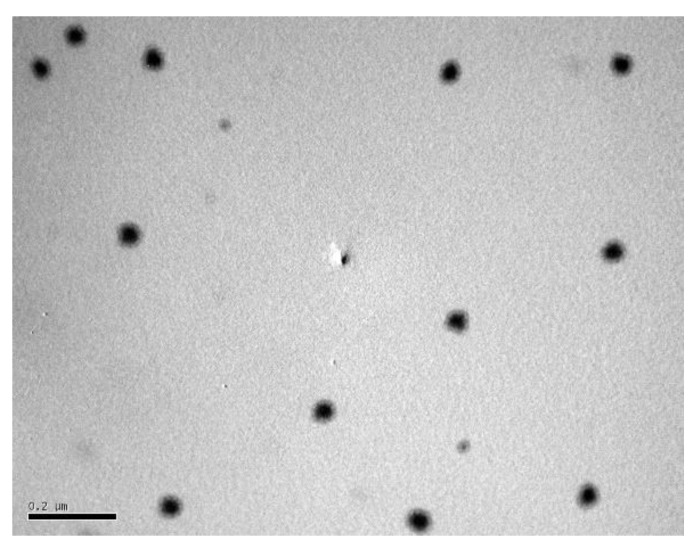
Transmission electron microscopy photo of TTO-SLN1.

**Figure 4 molecules-27-06143-f004:**
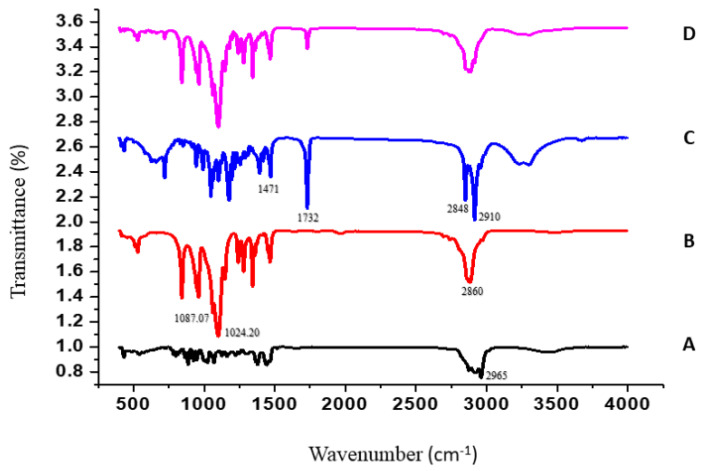
The FTIR spectra of TTO (**A**), P188 (**B**), GMS (**C**), and TTO-SLN1 (**D**).

**Figure 5 molecules-27-06143-f005:**
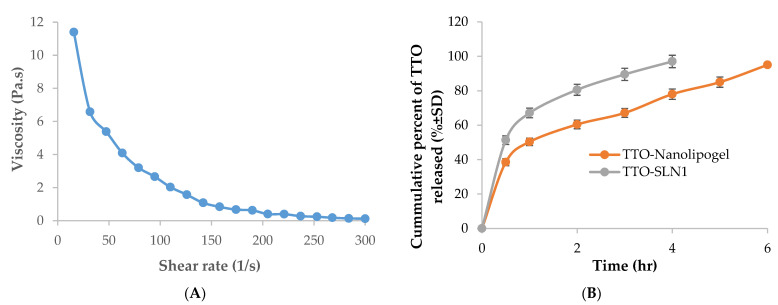
(**A**): Rheogram of TTO-loaded nanolipogel, (**B**): The in vitro release profile from TTO-nanolipogel was compared to that of TTO-SLN1.

**Figure 6 molecules-27-06143-f006:**
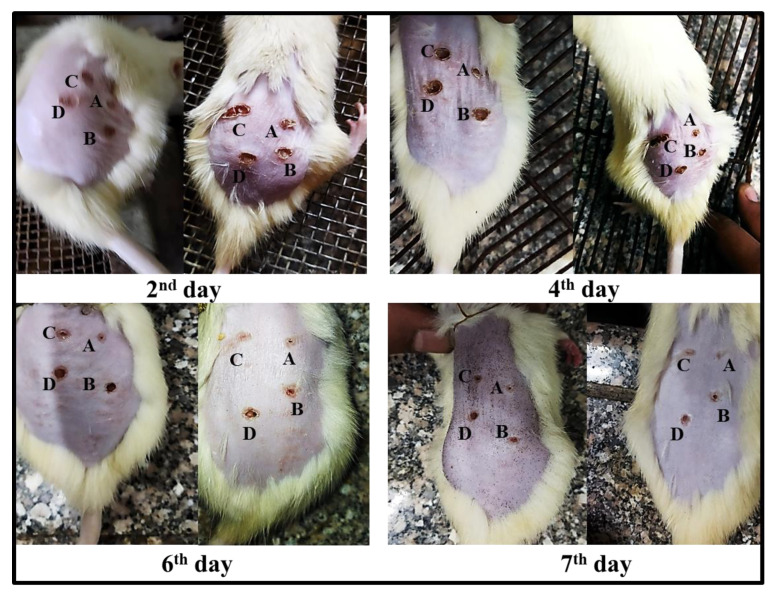
Burn healing process of two representative rats at different time intervals. Treatment was performed with the TTO-loaded nanolipogel (area A), the unmedicated nanolipogel (area B), TTO-loaded conventional gel (area C), and normal saline (area D).

**Figure 7 molecules-27-06143-f007:**
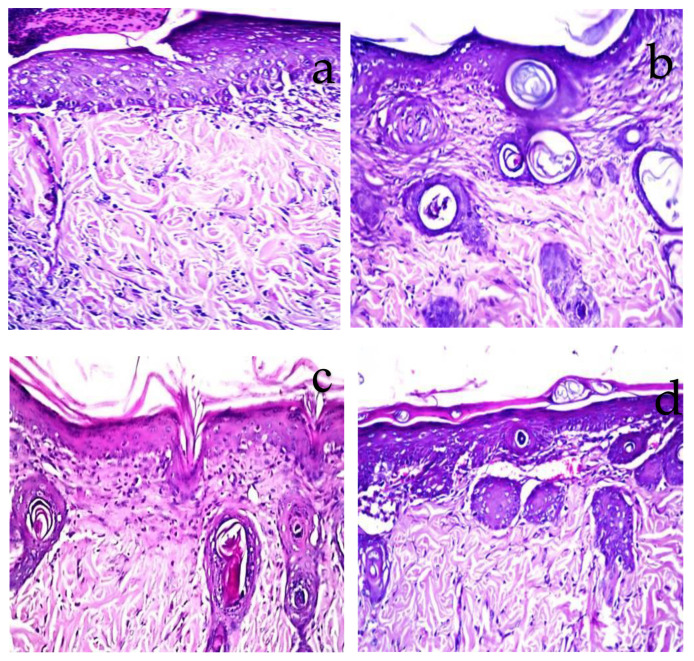
Photomicrographs of treated rat skin compared to the negative control taken at magnification ×40. (**a**) Skin area treated with normal saline showing oedema with focal ulceration and adjacent acanthosis in the epidermis with underlying granulation tissue and inflammatory cell infiltration in the dermis. (**b**) Skin area treated with TTO-loaded conventional gel showing few inflammatory cells infiltration in the dermis. (**c**) Skin of rat treated with the unmedicated nanolipogel showing acanthosis in the epidermis associated with few inflammatory cells infiltration in the dermis. (**d**) Skin area treated with TTO-loaded nanolipogel showing normal histological structure.

**Figure 8 molecules-27-06143-f008:**
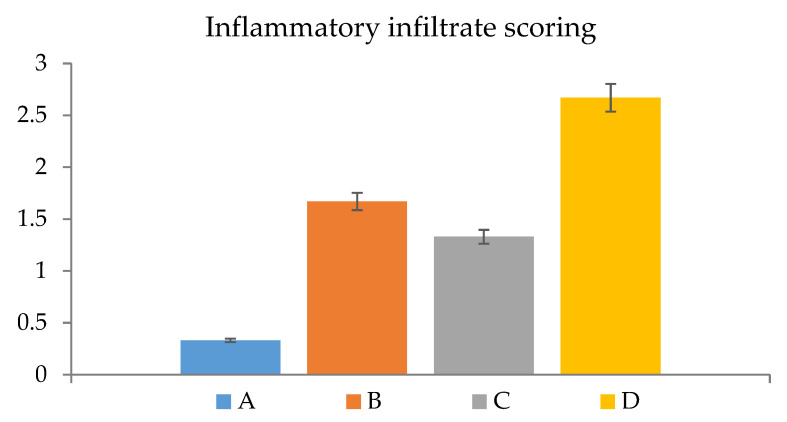
Semi-quantitative assessment of pathological changes after topical application of the different treatments (*n* = 5). TTO-loaded nanolipogel (A), the unmedicated nanolipogel (B), TTO-loaded conventional gel (C), and normal saline (D).

**Table 1 molecules-27-06143-t001:** Relative percentage of volatiles in tea tree oil specimen (*n* = 3) by GC-MS analysis.

No.	RT (min)	Compounds	KI	Class	Relative %
1	4.77	*γ*-Terpinene	983	Monoterpene hydrocarbons	2.88 ± 0.05
2	5.05	*α*-Terpinolene	1027	Monoterpene hydrocarbons	5.27 ± 0.84
3	6.8	Terpinen-4-ol	1112	Alcohol	4.19 ± 0.42
4	8.27	Terpinen-4-ol isomer	1139	Alcohol	7.87 ± 0.79
5	8.39	*α*-Terpineol	1148	Alcohol	4.43 ± 0.16
6	8.57	*α*-Cyclocitral	1164	Aldehyde	0.52 ± 0.04
7	8.74	7-Hydroxyterpineol	1210	Alcohol	1.61 ± 0.09
8	9.41	*trans*-Ascaridol glycol	1225	Alcohol	4.66 ± 0.43
9	9.58	*p*-Mentha-3-en-8-ol	1232	Alcohol	1.17 ± 0.05
10	10.04	*α*-Copaene	1241	Sesquiterpene hydrocarbons	1.7 ± 0.07
11	10.1	4-Heptenal	1265	Aldehyde	0.63 ± 0.03
12	10.43	Isoledene	1317	Sesquiterpene hydrocarbons	0.89 ± 0.07
13	10.53	*α*-Copaene isomer	1324	Sesquiterpene hydrocarbons	2.22 ± 0.16
14	10.84	*α*-Cadinol	1368	Alcohol	0.96 ± 0.31
15	11.17	*α*-Cadinol isomer	1383	Alcohol	2.84 ± 0.52
16	11.37	Caryophyllene	1462	Sesquiterpene hydrocarbons	2.58 ± 0.28
17	11.47	Caryophyllene isomer	1478	Sesquiterpene hydrocarbons	0.66 ± 0.05
18	11.87	*α*-Bergamotene	1493	Sesquiterpene hydrocarbons	5.3 ± 0.18
19	11.98	Alloaromadendrene	1529	Sesquiterpene hydrocarbons	1.5 ± 0.53
20	12.23	*δ*-Cadinene	1561	Sesquiterpene hydrocarbons	3.73 ± 0.59
21	12.43	*δ-*Cadinene isomer	1583	Sesquiterpene hydrocarbons	3.88 ± 0.76
22	13.05	*α*-Cubebene	1647	Sesquiterpene hydrocarbons	8.15 ± 2.37
23	13.6	*α*-Farnesene	1684	Sesquiterpene hydrocarbons	9.43 ± 3.62
24	13.68	*α*-Copaene isomer	1694	Sesquiterpene hydrocarbons	1.33 ± 0.06
25	13.77	*α*-Calacorene	1728	Aromatics	0.27 ± 0.09
26	14.06	10-Aromadendranol	1750	Sesquiterpene hydrocarbons	1.74 ± 0.16
27	14.18	Spathulenol	1765	Alcohol	2.25 ± 0.25
28	14.54	Unknown 1	1769	-	2.64 ± 0.04
29	14.76	Guaiol	1782	Alcohol	1.84 ± 0.08
30	14.88	Isospathulenol	1798	Alcohol	1.79 ± 0.03
31	15.26	7-epi-*α*-Eudesmol	1801	Alcohol	1.71 ± 0.29
32	15.35	Cubenol	1816	Alcohol	2.06 ± 0.45
33	15.61	Spathulenol isomer	1820	Alcohol	3.65 ± 0.82
34	15.92	Epoxyguaiene	1835	Sesquiterpene hydrocarbons	0.45 ± 0.27
35	21.68	Unknown 2	1841	-	1.08 ± 0.09
36	23.11	Unknown 3	1867	-	0.62 ± 0.37
37	35.02	Unknown 4	1886	-	0.75 ± 0.08
38	35.12	Unknown 5	1894	-	0.72 ± 0.05

**Table 2 molecules-27-06143-t002:** Physicochemical properties of TTO-SLNs (entrapment efficiency, loading capacity, particle size, zeta potential, and polydispersity index).

Formulations	Entrapment Efficiency (E.E. ± SD) %	TTO Loading (LC ± SD)%	Particle Size (P.S ± SD) nm	Polydispersity Index (PDI ± SD)	Zeta Potential (Z.P.) mV
**TTO-SLN1**	96.26 ± 2.3	29.81 ± 1.2	235.0 ± 20.4	0.31 ± 0.01	−32.0
**TTO-SLN2**	96.73 ± 2.5	19.63 ± 1.8	226.6 ± 20.6	0.34 ± 0.01	−29.3
**TTO-SLN3**	97.12 ± 3.1	21.40 ± 2.1	335.6 ± 19.7	0.38 ± 0.02	−25.4
**TTO-SLN4**	97.70 ± 3.0	15.23 ± 2.2	324.5 ± 33.4	0.37 ± 0.01	−26.5
**TTO-SLN5**	98.50 ± 2.8	14.31 ± 2.7	426.0 ± 40.7	0.44 ± 0.02	−25.0
**TTO-SLN6**	98.60 ± 3.0	10.23 ± 1.3	429.8 ± 35.8	0.43 ± 0.02	−26.6

**Table 3 molecules-27-06143-t003:** Percentage of burn area contraction after treatment with TTO-loaded nanolipogel (area A), the unmedicated nanolipogel (area B), TTO-loaded conventional gel (area C), and normal saline (area D).

Rat Number	Percentage of Burn Area Contraction											
	A			B			C			D		
	Day 4	Day 6	Day 7	Day 4	Day 6	Day 7	Day 4	Day 6	Day 7	Day 4	Day 6	Day 7
**1**	4.6	88.6	92.5	5.0	59.9	68.5	3.3	84.9	90.5	2.0	51.5	54.8
**2**	5.4	89.1	97.3	3.2	47.2	67.4	2.8	79.8	80.9	3.0	51.1	59.6
**3**	5.1	81.6	97.8	3.1	65.1	65.4	4.9	60.0	78.2	2.2	39.0	39.7
**4**	4.9	91.2	91.5	3.6	56.1	56.1	3.3	72.9	82.9	3.1	52.7	56.4
**5**	5.0	92.1	91.1	4.9	64.0	66.5	3.4	75.0	68.1	2.3	53.4	56.7
**Mean**	5.0	88.5	94.0	4.0	58.5	64.8	3.5	74.5	80.1	2.5	49.5	53.4
**SD**	0.3	4.1	3.2	0.9	7.2	5.0	0.8	9.3	8.1	0.5	6.0	7.9

**Table 4 molecules-27-06143-t004:** Composition of TTO-loaded SLNs.

Formulations	Composition	
	GMS (%*w/w*)	P188 (%*w/w*)
**TTO-SLN1**	2	5
**TTO-SLN2**	2	10
**TTO-SLN3**	4	5
**TTO-SLN4**	4	10
**TTO-SLN5**	8	5
**TTO-SLN6**	8	10

## Data Availability

All data are available in the text.

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
