# Peer review of "Nanolipogel Loaded with Tea Tree Oil for the Management of Burn: GC-MS Analysis, In Vitro and In Vivo Evaluation"

_molecules, 2022, doi:10.3390/molecules27196143_

Round 1

Reviewer 1 Report

see in the attached file

Author Response

We would like to thank the Editor and the reviewer for evaluating our manuscript. We very much appreciate the Reviewer’s comments and the recommendations to improve the manuscript.

All issues indicated in the review report have been addressed and queries are answered. We hope that this revised version can comply with the journal scope and requirements.

Thanks and Regards.

Reviewer #1

 In this manuscript authors describes the SLN loaded with tea tree oil for the repair of skin damage. The research also includes in vivo animal experiments, which is especially valuable. For this reason (the use of animals) they should be described as well as possible, so that the presented results are valuable and bring new knowledge in relation to the experience already possessed. Please find below my comments to the manuscript. Explanations to the following issues should be completed in selected parts of the manuscript.

  1. Abstract: Too long sentences make understanding and context difficult. Especially the abstract (the third sentence takes five lines) requires re-editing, and the paper should be corrected in this respect.

Manuscript was revised and sentences have been reformulated

  1. Section 3.2.9: It is mentioned (line 53) that the application of the plain oil may cause skin irritation. So why did the study only evaluate inflammation (3.2.9) and not irritation? Was inflammation testing performed according to generally accepted procedures? - there is no reference to literature in the methods (3.2.9).!

Generally, the irritation evaluation is done on intact skin. The examination of the safety and non-irritancy of the designed nanogel to the skin was performed and added to the manuscript.

Inflammation testing was performed according to a previously reported procedure by Lin etal., (Lin, Lee et al. 2015) . Reference is added in the text.

  1. Introduction line 55-56: "Also, the lipophilic nature of the oil can decrease its efficacy by diminishing its absorption through biological membranes and patient acceptability ..." with regard to this statement, there is no information about the possible mechanism of action of numerous active compounds from TTO (numerous mentioned and marked in the work), with particular emphasis on the site of action = structures in the skin that are their target site, especially since it is damaged skin. This issue should be discussed

More details related to this issue are added

[Hydrogel containing TTO reduced skin damage induced by ultraviolet B exposure and the hydrogel reduced wound area compared to allantoin-loaded hydrogel, posing TTO as a possible candidate to control skin inflammation and wounds (Flores, de Lima et al. 2015). TTO also exhibited topical antimicrobial effect against various fungi, bacteria, and yeasts (Brady, Loughlin et al. 2006, Hammer 2015). TTO has more than 100 components, with monoterpene and sesquiterpene hydrocarbons along with their associated alcohols being abundant classes. TTO major constituents i.e., terpinen-4-ol, α-terpineol, terpinolene, and terpinene were separately correlated to anti-inflammatory and antibacterial effects. Terpinen-4-ol was the greatest contributor to antibacterial efficacy against Staphylococcus aureus and Propionibacterium acnes (Raman, Weir et al. 1995, Lee, Chen et al. 2013). The most abundant component of TTO, terpinen-4-ol, exerted significant anti-inflammatory action via inhibiting the formation of TNF-α (tumor necrosis factor-α), prostaglandin E2, IL-1 (interleukin-1), IL-8, and IL-10 (Hart, Brand et al. 2000). Further, the cell membrane of Botrytis cinerea was the primary target of terpinen-4-ol indicative of TTO antifungal mode of action causing cell membrane leakage (Yu, Wang et al. 2015)]

  1. Section 3.2.2.: At the time of preparation of the SLN, the TTO was dissolved in GMS (line 304-305). What was the solubility of TTO in GMS - should be determined, it is important from the point of view of incorporation in the lipid matrix

GMS was previously used as a lipid matrix to formulate SLN incorporating a similar lipophilic active agent of natural origin. This is added to the manuscript.

          Also, the rational for the selection of the GMS is added to the manuscript

  1. Section 3.2.2.: What is the thermal resistance of TTO and its components, how long has TTO been exposed to elevated temperatures (cooling was slow) at the time of SLN preparation, and whether this could affect the composition of the TTO and its biological activity. Are the biologically active substances contained in TTO not sensitive to the effects of ultrasounds (even 30 min!).

TTO can maintain its integrity at temperature up to 40ËšC (Xu, Wei et al. 2022). During the formulation of TTO loaded SLNs, TTO was dissolved in the melted GMS after cooling to 40°C following by sonication for 30 minutes (two cycles of 15 min and intervals of 15 seconds) (Fathi, Varshosaz et al. 2013, Laein, Khanzadi et al. 2022), and the TTO-SLNs was obtained by cooling at room temperature.

These details are added in the text.

  1. Section 3.2.2.: What was the batch size of the SLN?

The batch size of all the prepared SLNs was 25 g. This is added to the manuscript.

  1. Section 3.2.3.: The EE parameter was determined from the amount of TTO determined after filtration through a 0.2 μm membrane. How was the adsorption of TTO to the membrane assessed (TTO loss on the membrane during filtration)? A low TTO content in the filtrate does not necessarily indicate complete incorporation into the SLN.

To accurately investigate the amount of TTO encapsulated in the SLNs, qualitative analysis was done for the supernatant (for assessment of encapsulation efficiency) as well as the precipitate (for assessment of loading capacity).

The almost absence of TTO in the supernatant while its presence in the collected residue (SLN) can confirm its incorporation in the prepared nanoparticles.

More details are added to the manuscript.

Section 3.2.5.: How much SLN was in the bag during the release study. What was the release area through the membrane (the area of the membrane through which active substance permeated into the acceptor fluid)? Was this surface the same for SLN dispersion and plain oil?

The weight of the used SLN was equivalent to 1 mg oil, so it was dependent on the encapsulation efficiency of each formula (the SLN weight used around 0.5 g).

The same membrane length (i.e. same dimensions of the dialysis bag) was used for all tested samples (SLN as well as plain oil). The area of the membrane was about 12.5 cm2. This is added to the manuscript.

  1. Section 3.2.5.: "The cumulative percent of released active agents was plotted against time." What was 100% in the dissolution test - TTO mass in SLN ?, theoretical, practical, determined amount of active substances ???

In order to study the percent released of TTO from the prepared TTO-SLNs compared to the free TTO, the in-vitro release study was conducted and the tested samples were analyzed spectrophotometrically. The cumulative percent of released active agents was plotted against time. A complete release corresponds to the release of the total amount of active ingredient encapsulated in the formulation. This in-vitro study is generally followed in most pharmaceutical studies.

  1. Section 3.2.5. 3.: "CNF gel was mixed with the TTO-loaded SLN in a ratio of 1: 1" - I understand that CNF was mixed with the 2% SLN dispersion?, it should be explained

The selected TTO-SLN1 (composed of 2% w/w GMS and 5% w/w poloxamer 188 was mixed with the CNF gel in a ratio 1:1 (w/w). This was listed in the manuscript.

  1. Discussion: It should be Table 2 and not Table 3.

Corrected

12.Discussion: What justifies the choice of the fastest-releasing formulation (4 h, in the gel 6 h), if the application took place once a day. There is no discussion of the potential application frequency with regard to the release profiles

This issue was clarified in the manuscript accompanied by references.

For controlling burns, a rapid onset of action is needed, therefore the fast releasing formulation is selected. After active agents’ release to the skin surface, then an extended effect is expected due to the accumulation of the lipid-type nano-formulation in the skin. It was previously reported that lipid-based nanocarriers with a particle size range around the value listed above (table 2) was beneficial for skin-targeting permitting for the deposition of the active agent within the skin layers. A once-daily application is generally followed in such a type of nanocarriers. Moreover, to facilitate the process of application in addition to taking advantage of the cell regeneration and tissue healing properties of the CNF, the formula was incorporated into the gel.

  1. Figure 6: If the photos in figure 6 show two selected rats - the same rats all the time, it is not clear what the dark spots are visible all over the rat's body on the left side (in figure 6 after 7 days), where they came from, what they mean.

The photos displayed in figure 6 are those of two representative rats. The dark spots appearing in the original photos are resulting from debris in the cages and towel used while handling the animal.

Brady, A., R. Loughlin, D. Gilpin, P. Kearney and M. Tunney (2006). "In vitro activity of tea-tree oil against clinical skin isolates of meticillin-resistant and -sensitive Staphylococcus aureus and coagulase-negative staphylococci growing planktonically and as biofilms." Journal of Medical Microbiology 55(10): 1375-1380.

Fathi, M., J. Varshosaz, M. Mohebbi and F. Shahidi (2013). "Hesperetin-Loaded Solid Lipid Nanoparticles and Nanostructure Lipid Carriers for Food Fortification: Preparation, Characterization, and Modeling." Food and Bioprocess Technology 6(6): 1464-1475.

Flores, F. C., J. A. de Lima, C. R. da Silva, D. Benvegnú, J. Ferreira, M. E. Burger, R. C. R. Beck, C. M. B. Rolim, M. I. U. M. Rocha, M. L. da Veiga and C. D. B. da Silva (2015). "Hydrogels containing nanocapsules and nanoemulsions of tea tree oil provide antiedematogenic effect and improved skin wound healing." Journal of nanoscience and nanotechnology 15(1): 800-809.

Hammer, K. A. (2015). "Treatment of acne with tea tree oil (melaleuca) products: A review of efficacy, tolerability and potential modes of action." International Journal of Antimicrobial Agents 45(2): 106-110.

Hart, P. H., C. Brand, C. F. Carson, T. V. Riley, R. H. Prager and J. J. Finlay-Jones (2000). "Terpinen-4-ol, the main component of the essential oil of Melaleuca alternifolia (tea tree oil), suppresses inflammatory mediator production by activated human monocytes." Inflammation Research 49(11): 619-626.

Laein, S. S., S. Khanzadi, M. Hashemi, F. Gheybi and M. Azizzadeh (2022). "Peppermint essential oil-loaded solid lipid nanoparticle in gelatin coating: Characterization and antibacterial activity against foodborne pathogen inoculated on rainbow trout (Oncorhynchus mykiss) fillet during refrigerated storage." J Food Sci 87(7): 2920-2931.

Lee, C.-J., L.-W. Chen, L.-G. Chen, T.-L. Chang, C.-W. Huang, M.-C. Huang and C.-C. Wang (2013). "Correlations of the components of tea tree oil with its antibacterial effects and skin irritation." Journal of Food and Drug Analysis 21(2): 169-176.

Lin, H. C., H. S. Lee, T. S. Chiueh, Y. C. Lin, H. A. Lin, Y. C. Lin, T. L. Cha and E. Meng (2015). "Histopathological assessment of inflammation and expression of inflammatory markers in patients with ketamine-induced cystitis." Mol Med Rep 11(4): 2421-2428.

Raman, A., U. Weir and S. F. Bloomfield (1995). "Antimicrobial effects of tea-tree oil and its major components on Staphylococcus aureus, Staph. epidermidis and Propionibacterium acnes." Letters in Applied Microbiology 21(4): 242-245.

Xu, Y., Y. Wei, S. Jiang, F. Xu, H. Wang and X. Shao (2022). "Preparation and characterization of tea tree oil solid liposomes to control brown rot and improve quality in peach fruit." LWT 162: 113442.

Yu, D., J. Wang, X. Shao, F. Xu and H. Wang (2015). "Antifungal modes of action of tea tree oil and its two characteristic components against Botrytis cinerea." Journal of Applied Microbiology 119(5): 1253-1262.

Reviewer 2 Report

1. Some previous publication has proved the efficiency of lipid-type nano-carriers for dermal delivery and repair [17-22]. Explain the findings of reported literature.

2. How the 0.2 % TTO selected for formulation. How the author selected the used components of preparation.

3. THe rationale of used composition is not justified.

4. How much is the drug load.

5. For better understanding mark the characteristic peaks in IR.

6. All the formulations showed about 85 % EE, then what is the use of different compositions.

Author Response

Reviewer #2

  1. Some previous publication has proved the efficiency of lipid-type nano-carriers for dermal delivery and repair [17-22]. Explain the findings of reported literature.

More details are listed briefly.

These studies were focused on enhancing the solubility and efficacy of natural compounds with low aqueous solubility (such as rutin, diosmin, resveratrol, curcumin and lutein) to deliver them efficiently to the skin. The results proved the possible amelioration of their effects by nanoencapsulation in lipid-based colloidal carriers

  1. How the 0.2 % TTO selected for formulation. How the author selected the used components of preparation.

The concentration of the oil was selected according to the following studies

Guo et al,  (Guo, Ma et al. 2022) who an active ingredient (phytosterol) equivalent to 10 % of the used lipid (GMS) weight

Laein et al., (Laein, Khanzadi et al. 2022) who developed peppermint essential oil loaded solid lipid nanoparticles by incorporating 0.2% oil in 2% GMS

Also, the rational for the selection of the SLN components is added to the manuscript

  1. THe rationale of used composition is not justified.

Rational for the selection of the SLN components is added to the manuscript

  1. How much is the drug load?

The percent drug loading for all the prepared formulations are added in table 2.

  1. For better understanding mark the characteristic peaks in IR.

The characteristic peaks are marked in the IR spectra

  1. All the formulations showed about 85 % EE, then what is the use of different compositions.

The study comprised the evaluation of different physicochemical properties. The formulation composition may influence some of them, not necessarily all. As depicted from the results, the variation in the formulation composition exhibited no significant difference in the entrapment efficiencies. However, it showed a significant difference in the amount of TTO released and, leaded to the choice of the best formulation displaying the highest percent of active agent release.

Guo, S.-J., C.-G. Ma, Y.-Y. Hu, G. Bai, Z.-J. Song and X.-Q. Cao (2022). "Solid lipid nanoparticles for phytosterols delivery: The acyl chain number of the glyceride matrix affects the arrangement, stability, and release." Food Chemistry 394: 133412.

Laein, S. S., S. Khanzadi, M. Hashemi, F. Gheybi and M. Azizzadeh (2022). "Peppermint essential oil-loaded solid lipid nanoparticle in gelatin coating: Characterization and antibacterial activity against foodborne pathogen inoculated on rainbow trout (Oncorhynchus mykiss) fillet during refrigerated storage." J Food Sci 87(7): 2920-2931.